# Proteomic and Biochemical Approaches Elucidate the Role of Millimeter-Wave Irradiation in Wheat Growth under Flooding Stress

**DOI:** 10.3390/ijms231810360

**Published:** 2022-09-08

**Authors:** Setsuko Komatsu, Yoshie Tsutsui, Takashi Furuya, Hisateru Yamaguchi, Keisuke Hitachi, Kunihiro Tsuchida, Masahiko Tani

**Affiliations:** 1Faculty of Environment and Information Sciences, Fukui University of Technology, Fukui 910-8505, Japan; 2Research Center for Development of Far-Infrared Region, University of Fukui, Fukui 910-8507, Japan; 3Department of Medical Technology, Yokkaichi Nursing and Medical Care University, Yokkaichi 512-8045, Japan; 4Institute for Comprehensive Medical Science, Fujita Health University, Toyoake 470-1192, Japan

**Keywords:** proteomics, wheat, flooding, millimeter-wave irradiation

## Abstract

Flooding impairs wheat growth and considerably affects yield productivity worldwide. On the other hand, irradiation with millimeter waves enhanced the growth of chickpea and soybean under flooding stress. In the current work, millimeter-wave irradiation notably enhanced wheat growth, even under flooding stress. To explore the protective mechanisms of millimeter-wave irradiation on wheat under flooding, quantitative proteomics was performed. According to functional categorization, proteins whose abundances were changed significantly with and without irradiation under flooding stress were correlated to glycolysis, reactive-oxygen species scavenging, cell organization, and hormonal metabolism. Immunoblot analysis confirmed that fructose-bisphosphate aldolase and β tubulin accumulated in root and leaf under flooding; however, even in such condition, their accumulations were recovered to the control level in irradiated wheat. The abundance of ascorbate peroxidase increased in leaf under flooding and recovered to the control level in irradiated wheat. Because the abundance of auxin-related proteins changed with millimeter-wave irradiation, auxin was applied to wheat under flooding, resulting in the application of auxin improving its growth, even in such condition. These results suggest that millimeter-wave irradiation on wheat seeds improves the recovery of plant growth from flooding via the regulation of glycolysis, reactive-oxygen species scavenging, and cell organization. Additionally, millimeter-wave irradiation could promote tolerance against flooding through the regulation of auxin contents in wheat.

## 1. Introduction

Wheat is an important staple crop, and its availability affects the livelihoods of nearly every family in the world [1]. An estimated 15–20% of its cultivating area faces flooding each year [2,3], resulting in waterlogging reducing average grain yield by 43% [4]. In addition to climate changes, flooding is expected to increasingly cause severe crop declines in terms of both yield and quality worldwide [5]. Flooding is classified into two forms depending on water depth, which are submergence and waterlogging [6]. Submergence is the condition in which the whole plant is completely or partially immersed in water, while waterlogging is the state where water exists on the soil surface and crop roots are surrounded by water [3]. Flooding threatens to drastically reduce crop yield and is expected to be even more severe in many parts of the world due to climatic anomalies in the future [7]. Understanding the mechanisms of wheat coping with unexpected flooding is important for developing new flooding-tolerant wheat cultivars.

In the electromagnetic spectrum, millimeter waves lie at the intersection of microwaves and infrared. The radio frequency of millimeter waves extends from 30 to 300 GHz, corresponding to wavelengths from 10 to 1 mm [8]. Millimeter-wave irradiation with long wavelengths and minimal risks to human health is an appropriate technology for the environment, which have dynamic effects on organisms [9]. Wheat seeds treated with millimeter waves grew into higher plants with better biological harvest compared to those without treatment [10]. Millimeter-wave irradiation on wheat seed at the initial stage improved not only seed germination [11,12], but also shoot growth and grain yield [13]. On the other hand, millimeter-wave irradiation improved the growth of seedling and the flooding tolerance of soybean [14], and chickpea [15]. These findings indicated that the millimeter-wave irradiation may be an effective approach to promote plant growth and stress tolerance in crops; however, the effects on wheat growth under flooding have not been investigated.

In brown rice, millimeter-wave irradiation, which improved germination ratio, increased polyphenol content and decreased γ-aminobutyric acid [16]. In soybean, millimeter-wave irradiation promoted the recovery of seedlings under oxidative stress, which positively regulated glycolysis and redox-related pathways [14]. In chickpea, millimeter-wave irradiation promoted the recovery of plant growth under flooding by regulating photosynthesis in leaf and fermentation in roots via cell death [15]. To determine the effects of millimeter-wave irradiation on wheat growth under flooding stress, which is submergence stress, morphological analysis was performed through comparisons among treatments in different ranges of dose and various durations of irradiation. Based on the morphological results, proteomic analysis was carried out to explore the responsible mechanisms for the positive effects of millimeter waves on wheat growth under flooding. Proteomic results were subsequently confirmed by immunoblot, enzyme activity, and physiological analyses.

## 2. Results

### 2.1. Morphological Changes of Wheat Irradiated with Millimeter Waves under Flooding Stress

To clarify the effect of millimeter-wave irradiation on wheat under flooding stress, morphological changes of wheat treated with a variety of irradiation were analyzed. Wheat seeds were irradiated with 0, 10, 20, and 40 mW of millimeter waves for 0, 10, 20, and 40 min using a Gunn oscillator (Appendix A). Without flooding stress, the length and weight of root and leaf of wheat did not change significantly (Appendix A). Under flooding stress, the duration of 20 min was most effective in promoting wheat growth (Figure 1). Additionally, wheat growth was improved with 20 mW millimeter-wave irradiation (Figure 1). Although the growth of wheat seedlings was suppressed by flooding (Figure 1) compared with non-flooding conditions (Appendix A), its growth was enhanced by the irradiation of 20 mW of millimeter waves for 20 min, even under flooding conditions. (Figure 1). These results indicated that the irradiation of 20 mW of millimeter waves for 20 min was more effective in root growth. Based on morphological results, the condition of 20 mW of millimeter-wave irradiation for 20 min was used for the seed treatment for proteomic analysis.

### 2.2. Protein Identification and Functional Categorization in Wheat Irradiated with Millimeter Waves under Flooding Stress

To clarify the cellular mechanism on plant growth of wheat seeds irradiated with millimeter waves, a gel- and label-free proteomics was conducted using root (Appendix A). Four kinds of treatments, which are irradiated/unirradiated and flooding/non-flooding, were performed (Appendix A). Proteins were extracted from root, whose seeds were irradiated with or without 20 mW of millimeter waves for 20 min, under non-flooded or flooded conditions. Proteins extracted were enriched, reduced, alkylated, digested, and analyzed using nano-liquid chromatography (LC) combined with mass spectrometry (MS). The relative protein abundance of irradiated wheat was compared with that of unirradiated wheat under non-flooding (Appendix A) or flooding (Appendix A) conditions. Furthermore, relative protein abundance of non-flooded wheat was compared with that of flooded wheat with unirradiation (Appendix A) or irradiation (Appendix A).

The abundance of 254 and 784 proteins differentially changed with fold change ≥1.5 and ≤2/3 in the roots of millimeter-wave irradiated wheat compared with unirradiated wheat under non-flooding condition and flooding condition, respectively (Appendix A). Functional category of identified proteins was obtained using MapMan bin codes (Figure 2). Among the 254 proteins, 165 proteins increased, and 89 proteins decreased with irradiation compared with unirradiation under non-flooding (Figure 2 left). This indicated that the proteins were not largely changed between irradiation and unirradiation without flooding stress. Among the 784 proteins, 361 proteins increased, and 423 proteins decreased with irradiation compared with unirradiation under flooding (Figure 2 right). Compared to unirradiation under flooding, irradiation significantly changed functional categories, which were glycolysis, minor CHO, hormonal metabolism, and redox, which is reactive-oxygen species (ROS) scavenging.

Furthermore, the abundance of 2678 and 704 proteins differentially changed with fold change ≥1.5 and ≤2/3 in the roots of millimeter-wave unirradiated wheat and irradiated wheat, respectively, under flooding conditions compared with under non-flooding conditions (Appendix A). Among the 2678 proteins, 1609 proteins increased, and 1069 proteins decreased without irradiation under flooding stress compared with non-flooding condition (Figure 3 left). Among the 704 proteins, 425 proteins increased, and 279 proteins decreased with irradiation under flooding stress compared with non-flooding condition (Figure 3 right). Functional category of identified proteins was obtained using MapMan bin codes (Figure 3). Cell organization and hormonal metabolism were oppositely changed between irradiation and unirradiation under flooding conditions. Based on proteomic results, proteins related to glycolysis, ROS scavenging, cell organization, and hormonal metabolism were further confirmed using biochemical and physiological techniques.

### 2.3. Activity of Alcohol Dehydrogenase (ADH) in Wheat Irradiated with Millimeter Waves under Flooding Stress

Because the abundance of ADH was not clearly changed in wheat with millimeter wave-irradiation under flooding based on proteomic results (Figure 2 and Figure 3), ADH activity was analyzed (Figure 4). Proteins were extracted from root and leaf of wheat, whose seeds were irradiated with or without 20 mW of millimeter waves for 20 min, under non-flooded or flooded conditions. Although ADH activity was markedly increased in root and leaf of wheat under flooding stress, its activity was the same in wheat with and without millimeter-wave irradiation (Figure 4). These results indicated that the fermentation increased by flooding was not changed with millimeter-wave irradiation.

### 2.4. Abundance of Proteins Related to Glycolysis in Wheat Irradiated with Millimeter Waves under Flooding Stress

To further reveal the change of accumulation of proteins from various treatments, immunoblot analysis of proteins related to glycolysis was carried out (Figure 5). Proteins were extracted from root and leaf of wheat, whose seeds were irradiated with or without 20 mW of millimeter waves for 20 min, under non-flooded or flooded conditions. The staining pattern of Coomassie-brilliant blue was used as a loading control (Appendix A). For confirmation of the change of glycolysis related proteins, the abundance of fructose-bisphosphate aldolase (FBA), triose-phosphate isomerase (TPI), and glyceraldehyde-3-phosphate dehydrogenase (GAPDH) was analyzed using immunoblot analysis (Appendix A). FBA accumulated in root and leaf under flooding; however, its accumulation was recovered to control level in irradiated wheat, even if it was this condition (Figure 5). The abundance of GAPDH increased in root and decreased in leaf of irradiated wheat under flooding (Figure 5). The abundance of TPI was not changed with any treatments (Figure 5). These results indicated that glycolysis was improved by millimeter-wave irradiation, even under flooding condition.

### 2.5. Abundance of Proteins Related to ROS Scavenging in Wheat Irradiated with Millimeter Waves under Flooding Stress

To further reveal the change of accumulation of proteins from various treatments, immunoblot analysis of proteins related to ROS scavenging was carried out (Figure 6). For confirmation of the change of ROS scavenging related proteins, the abundance of ascorbate peroxidase (APX), glutathione reductase (GR), and peroxiredoxin (PRX) was analyzed using immunoblot analysis (Appendix A). The abundances of APX, GR, and PRX increased in root under flooding stress; however, their abundances did not change with and without millimeter-wave irradiation (Figure 6). The abundance of APX increased in leaves under flooding stress and its accumulation was recovered to control level in irradiated wheat, even under flooding conditions (Figure 6). These results indicated that ROS scavenging was improved by millimeter-wave irradiation, even under flooding conditions.

### 2.6. Abundance of Proteins Related to Cell Organization in Wheat Irradiated with Millimeter Waves under Flooding Stress

To further reveal the change of accumulation of proteins from various treatments, immunoblot analysis of proteins related to cell organization was carried out (Figure 7). For confirmation of the change of cell organization related proteins, the abundance of β actin and β tubulin was analyzed using immunoblot analysis (Appendix A). The abundances of β actin increased in root under flooding stress; however, their abundances did not change with and without millimeter-wave irradiation (Figure 7). β tubulin accumulated in root and leaf under flooding; however, its accumulation was recovered to control level in irradiated wheat, even under this condition (Figure 7). These results indicated that cell organization was improved by millimeter-wave irradiation, even under flooding conditions.

### 2.7. Morphological Changes of Wheat after Auxin Application under Flooding Stress

Because protein abundance related to the metabolism of phytohormone, which is auxin, was changed by millimeter-wave irradiation (Figure 2 and Figure 3), morphological changes of wheat treated with a variety of treatments were analyzed (Figure 8). The length/fresh weight of leaf/root of wheat were suppressed under flooding condition; however, they were improved with millimeter-wave irradiation, even under flooding condition (Figure 8). Furthermore, wheat growth was improved by the application of auxin under flooding stress compared with flooding only (Figure 8). This result indicated that auxin, among phytohormone, was one of the candidates for escaping from flooding stress. 

## 3. Discussion

### 3.1. Millimeter-Wave Irradiation Has a Positive Effect on Wheat Growth under Flooding Stress

Due to the characteristics of millimeter-wave irradiation, it is an environmentally compatible technology with minimal risks to human health, which is important for sustainable development and deserves research on its impact [17]. The effective mechanism of millimeter waves was the induction of thermal energy into the biological system through incident irradiation, resulting in localized heating of water molecules on the surface of cell membranes [18]. Additionally, several non-thermal effects of millimeter-wave irradiation were discovered, revealing that optimum millimeter-wave irradiation stimulated cell division, enzyme synthesis, growth rate, and biomass yield of a variety of micro-organisms [19]. These physiological and biochemical effects of millimeter-wave irradiation on micro-organisms facilitate the investigation of wheat irradiation for improving productivity in the agricultural industry. 

In this study, the Japanese bread wheat cultivar Nourin 61 was used. Nourin 61 is a representative Japanese cultivar of bread wheat, which is characterized by broad adaptation and environmental robustness [20]. It was utilized for a broad range of physiological and molecular studies, such as mutant screening and transgenic experiments [21]. It is indicated that the features of Nourin 61 might be established as the reference genotype for adaptation and breeding research. In this study, this wheat seeds were irradiated with millimeter waves, to determine the effect of millimeter-wave irradiation on wheat growth under flooding stress. Millimeter-wave irradiation significantly improved wheat growth, even under flooding (Figure 1). Millimeter-wave irradiation improved the growth of seedling and the flooding tolerance of soybean [14], and chickpea [15]. The morphological effects on wheat introduced by millimeter-wave irradiation are similar to the cases of soybean and chickpea. Current results with previous findings indicate that the irradiation of millimeter waves may be a potential tool to ensure wheat growth under flood stress.

### 3.2. Millimeter-Wave Irradiation Suppresses Glycolysis in Wheat under Flooding Stress

Oxygen deficiency in plant cells leads to the enhancement of metabolic processes such as sucrose catabolism, glycolysis, and fermentation pathways, which are important for energy conservation [22,23]. In flooding conditions, the energy to maintain plant vitality mainly relies on the ethanol-metabolic pathway in glycolysis to degrade glucose and glycogen accompanied by ATP generation [24]. Under flooding stress, a plant-derived smoke solution enhanced wheat growth; and FBA/GAPDH among glycolysis, which increased under flooding, decreased with its application under the same condition [25]. In this study, FBA was accumulated in root/leaf under flooding; and its accumulation was recovered to the control level in irradiated wheat (Figure 5). Furthermore, GAPDH was suppressed in wheat by millimeter-wave irradiation, even under flooding conditions (Figure 5). Millimeter-wave irradiation as well as the application of plant-derived smoke solution suppresses the glycolysis pathway in wheat, which could mildly generate the energy for surviving for a long-term period under flooding stress.

ADH is a key enzyme in the ethanol-fermentation pathway and subsequently in the adaptive anaerobic metabolism of plant tissue [24]. ADH activity significantly increased and mildly increased in the wild type and flooding-tolerant mutant soybeans, respectively, under flooding conditions compared with the non-flooding [26]. Furthermore, the application of silver nanoparticles enhanced flooding tolerance in soybeans; and fermentation-related proteins, which increased under flooding, decreased in response to silver nanoparticles, even under flooding [27]. The flooding tolerance of plants was proportional to the change in ADH activity in response to flooding. In chickpeas, ADH accumulation and activity increased under flooding; however, they were recovered with millimeter-wave irradiation through the formation of lateral roots [15]. On the other hand, fermentation-related proteins did not change in irradiated soybeans compared with unirradiated soybeans under flooding conditions [14]. Present results also indicated that fermentation was not changed with millimeter-wave irradiation (Figure 4). This result in previous reports suggests that wheat irradiated with millimeter waves might have other mechanisms to survive flooding stress compared with chickpeas.

### 3.3. Millimeter-Wave Irradiation Suppresses ROS Scavenging and Cell Organization in Wheat under Flooding Stress

Flooding stress triggered the production of ROS [28], which increased cell-membrane permeability, lipid peroxidation, and electrolyte leakage [29]; however, the enzymatic defense system composed of different antioxidant enzymes. The activities of several ROS-scavenging enzymes, including catalase, APX, GR, and superoxide dismutase, increased during flooding stress [30]. Meanwhile, ROS accumulation triggers the expression of downstream genes of fermentation required for hypoxia acclimation and survival [31]. For example, compared with waterlogging-tolerant wheat cultivar, the gene expressions and activities of catalase and Mn-superoxide dismutase decreased in waterlogging-susceptible wheat cultivar [32]. In this study, among ROS-scavenging enzymes, APX increased in leaf under flooding and recovered to the control level in irradiated wheat (Figure 6). On the other hand, in wheat leaf, the abundance of GR and PRX did not change with flooding stress (Figure 6). In wheat root, the abundance of APX, GR, and PRX increased under flooding stress; however, this accumulation did not recover with millimeter-wave irradiation. These results indicate that APX via the regulation of ROS production could be considered an important component of adaptive responses to flooding for wheat irradiated with millimeter waves compared with other ROS-scavenging enzymes.

ROS act as signaling molecules during plant-cell division, but their imbalance affects the tubulin cytoskeleton of dividing root cells of wheat and *A. thaliana* [33]. Experimental disturbance of ROS homeostasis rapidly stimulates microtubule disruption, leading to the assembly of resistant atypical tubulin polymers, microtubules, and tubulin paracrystals [34]. ROS imbalance is a stressful condition for plant cells with dramatic changes in the tubulin cytoskeleton [35]. Microtubules are regarded as emerging components of sensory mechanisms in plants, as various types of stress induce reorganization of the microtubule cytoskeleton [36]. In this study, β tubulin among cell-organization-related proteins accumulated in root/leaf under flooding; and its accumulation was recovered to the control level in irradiated wheat, even under flooding conditions (Figure 7). This present result is consistent with previous findings which suggest that β tubulin might function through ROS signaling during the early stage of wheat growth. 

### 3.4. Millimeter-Wave Irradiation Regulates Auxin Metabolism in Wheat under Flooding Stress

Phytohormones were demonstrated to play significant roles in flooding stress (7). Among them, ethylene is a primary signaling molecule, which is accumulated in plants to adapt to flooding stress [37]. Auxin, abscisic acid, gibberellic acid, cytokinin, and salicylic acid protected plants against flooding stress by regulating adventitious root formation or by controlling carbohydrate consumption [38]. Additionally, ethylene interacted with a hormonal cascade of auxin, abscisic acid, and gibberellic acid to promote adventitious root growth upon flooding in rice, tomato, and bitterzoet [39]. Application of spermidine alleviated plant-growth inhibition and reduced oxidative damage from hypoxic stress [40]. Spermidine upregulated the expression of auxin-related genes, which were auxin responsive factor1 and auxin1/auxin2/auxin3/auxin4 proteins; but downregulated the expression of ethylene-related 1-aminocyclopropane-1-carboxylic acid oxidase and synthase genes during flooding [40].

Abscisic acid and ethylene responses were activated under both submergence and drought stresses, whereas the auxin response was stimulated under submergence-specific stress; suggesting that auxin may be a signaling component, which distinguishes submergence-specific regulation [41]. Additionally, root growth cessation via ethylene and auxin rapidly occurred; and this quiescence behavior contributed to enhance hypoxia tolerance [42]. On the other hand, out of four auxins, which are indole-3-acetic acid, indole-3-propionic acid, indole-3-butyric acid, and 1-naphthylacetic acid, treatment with indole-3-butyric acid resulted in a high rooting rate and beneficial root morphology [43]. Additionally, 1.5 mg/L indole-3-butyric acid exhibited the highest rooting responses and 1.0 mg/L indole-3-butyric acid improved fruit yield and biomass [44]. In this study, auxin-metabolism-related proteins were identified with the irradiation of millimeter waves; and the application of auxin improved wheat growth under flooding conditions although its growth was suppressed by flooding (Figure 8). These results are consistent with previous reports which indicate that millimeter-wave irradiation might promote wheat tolerance under flooding through the regulation of auxin contents.

## 4. Materials and Methods

### 4.1. Plant Material, Millimeter-Wave Irradiation, and Treatment

As a millimeter-wave source, a Gunn oscillator (J. E. Caristrom, Chicago, IL, USA) was used as a millimeter-wave source (Appendix A). The frequency range of the Gunn oscillator is 79 to 115 GHz. The output power is 7 to 80 mW, depending on the output frequency. The Gunn oscillator was used in free running mode at 110 GHz. The electromagnetic waves emitted from the Gunn oscillator pass through an isolator, after adjusting the output power by an attenuator, then the electromagnetic wave is output to the free space via the horn antenna. The antenna pattern of the horn antenna has an aperture angle of 17 degrees on each side [14,15]. By placing a 5 cm diameter petri dish containing the seeds of wheat (*Triticum aestivum* L. cultivar Nourin 61) at 15 cm from the horn antenna, the millimeter-wave radiation area fully covers the dish. To investigate the dependence of millimeter-wave irradiation on intensity, the irradiation time was fixed at 20 min and 4 patterns of oscillation power were used: 0, 10, 20, and 40 mW. The average intensity of the electromagnetic waves irradiated to the seeds is 0, 0.13, 0.25, and 0.51 mW/cm^2^, respectively. The irradiated electromagnetic waves have a Gaussian distribution, with the maximum intensity at the center of the beam being 2.38 times of the average intensity, and the minimum intensity at the rim of the beam being 0.32 times of the average intensity. For the investigation of the irradiation-time dependence, the irradiation was performed with a fixed power of 20 mW for 0, 10, 20, and 40 min. The temperature rise of wheat was estimated to be well below 1 K with even the maximum-irradiation intensity and irradiation-time.

After irradiation, seeds were sterilized with 2% sodium hypochlorite solution, rinsed twice in water, and sown in 400 mL of silica sand in a seedling case. A total of 20 seeds were sown evenly in each seedling case. Wheats were grown at 25 °C and 60% humidity under white fluorescent light (160 µmol m^−2^ s^−1^, 16 h light period/day). To induce flooding stress, water was added to 5 cm above the sand surface to immerse 3-day-old wheats for 3 days. Irradiated/unirradiated and flooded/non-flooded wheats were collected. For morphological analysis, roots, and leaves of 6-day-old wheats were collected. For proteomic analysis, roots of 6-day-old wheats were collected. Three independent experiments were performed as biological replications for all experiments, meaning that the seeds were sown on different days (Appendix A).

### 4.2. Protein Extraction

A portion (500 mg) of samples was clipped into small pieces and put into a mortar and pestle. It was ground in 500 µL of lysis buffer, which contains 7 M urea, 2 M thiourea, 5% CHAPS, and 2 mM tributylphosphine. The suspension was centrifuged twice with 16,000× *g* for 10 min at 4 °C. The detergents from the supernatant were removed using the Pierce Detergent Removal Spin Column (Pierce Biotechnology, Rockford, IL, USA). The protein concentration was determined with bovine-serum albumin as the standard [45].

### 4.3. Protein Enrichment, Reduction, Alkylation, and Digestion

Extracted proteins (100 μg) were adjusted to a final volume of 100 μL. The protocol of protein enrichment, reduction, alkylation, and digestion are described in the previous study [46] (Appendix A).

### 4.4. Protein Identification Using Nano-Liquid Chromatography (LC) Mass Spectrometry (MS)

The LC (EASY-nLC 1000; Thermo Fisher Scientific, San Jose, CA, USA) conditions as well as the MS (Orbitrap Fusion ETD MS; Thermo Fisher Scientific) acquisition settings are described in the previous study [47] (Appendix A).

### 4.5. MS Data Analysis

The MS/MS searches were carried out using MASCOT (version 2.6.1, Matrix Science, London, UK) and SEQUEST HT search algorithms against the *Arabidopsis Thaliana* (UniProtKB TaxID = 3702) (version 2021-02) and *Triticum aestivum* (SwissProt TaxID = 4565) (version 2021-02) using Proteome Discoverer (version 2.4; Thermo Fisher Scientific, Waltham, MA, USA). The settings of MASCOT are described in the previous study [15] (Appendix A).

### 4.6. Differential Analysis of Proteins Using MS Data

Label-free quantification was also performed with Proteome Discoverer using precursor ions quantifiler nodes. For the differential analysis of the relative abundance of peptides and proteins between samples, the free software Perseus (version 1.6.15.0; Max Planck Institute of Biochemistry, Martinsried, Germany) [48] was used. Differential analysis is described in the previous study [15] (Appendix A). 

### 4.7. Immunoblot Analysis

Extracted proteins (10 μg) were adjusted to a final volume of 10 μL and mixed with SDS-sample buffer, which contains 60 mM Tris-HCl (pH 6.8), 2% SDS, 10% glycerol, and 5% dithiothreitol [49]. Proteins (10 µg) were separated by electrophoresis on a 10% SDS-polyacrylamide gel and transferred onto a polyvinylidene difluoride membrane using a semidry transfer blotter (Nippon Eido, Tokyo, Japan). The blotted membrane was blocked for 5 min in Bullet Blocking One regent (Nacalai Tesque, Kyoto, Japan). After blocking, the membrane was cross-reacted with a 1: 1000 dilution of the primary antibodies for 30 min at room temperature. As the primary antibodies, the following were used: anti-ascorbate peroxidase (APX) [50], glutathione reductase (GR) (Agrisera, Vännäs, Sweden), peroxiredoxin (PRX) [51], fructose-bisphosphate aldolase (FBA) [52], triose-phosphate isomerase (TPI) [14], glyceraldehyde-3-phosphate dehydrogenase (GAPDH) [14], β actin (Proteintech, Rosemont, IL, USA), and β tubulin (Proteintech) antibodies. As the secondary antibody, anti-rabbit IgG conjugated with horseradish peroxidase (Bio-Rad, Hercules, CA, USA) was used with 30 min incubation. After 30 min incubation, signals were detected using TMB Membrane Peroxidase Substrate Kit (Seracare, Milford, MA, USA). The integrated densities of bands were calculated using Image J software (version 1.53e with Java 1.8.0_172; National Institutes of Health, Bethesda, MD, USA). Coomassie brilliant blue staining was used as a loading control.

### 4.8. Measurement of Alcohol Dehydrogenase (ADH) Activity

ADH-activity assay was performed using Alcohol Dehydrogenase Activity Colorimetric Assay Kit (BioVision, Milpitas, CA, USA). A portion (50 mg) of samples was homogenized in 200 μL of ADH assay buffer and centrifuged at 13,000× *g* for 10 min at 4 °C to remove insoluble material. Extracts (50 μL) were added in 100 μL of reaction mixture containing 82 μL of ADH assay buffer, 10 μL of substrate, and 8 μL of developer. After mixing, the mixture was incubated for 2 and 10 min at 37 °C and the absorbance of mixture was measured at 450 nm.

### 4.9. Morphological Analysis after Auxin Application

As auxin, indole-3-butyric acid (Wako, Osaka, Japan) was used. For the auxin treated group, wheat seedlings were treated with 1 mg/L 4-(3-Indolyl) butyric acid under flooding stress for 3 days after 3-day germination and samples were collected. Leaf length, leaf-fresh weight, main-root length, and total-root weigh were measured as morphological parameters.

### 4.10. Statistical Analysis

Data were analyzed by one-way ANOVA followed by Tukey’s multiple comparison among multiple groups using SPSS (IBM, Chicago, IL, USA). The statistical significance of the 2 groups was evaluated by the Student’s *t*-test. A *p*-value of less than 0.05 was considered as statistically significant.

## 5. Conclusions

Millimeter-wave irradiation improved wheat growth [13], which is the most important staple crop, and its availability can impact the livelihoods of nearly every family globally. Additionally, irradiation with millimeter waves was a potential approach for promoting the flooding tolerance of soybeans [14], and chickpeas [15]. In this study, millimeter-wave irradiation also improved wheat growth, even under flooding stress. To clarify the dynamic effects of millimeter-wave irradiation on wheat under flooding, a gel- and label-free proteomic analysis and its confirmation analysis were conducted. The key findings were as follows: (i) FBA among glycolysis accumulated in root/leaf under flooding and its accumulation was recovered to the control level in irradiated wheat; (ii) APX among ROS scavenging increased in leaf under flooding and recovered to the control level in irradiated wheat; (iii) β tubulin among cell organization accumulated in root/leaf under flooding and its accumulation was recovered to the control level in irradiated wheat; and (iv) the application of auxin improved wheat growth under flooding conditions, although its growth was suppressed by flooding. These findings suggest that irradiation with millimeter waves on wheat seeds improves the recovery of plant growth from flooding through regulation of glycolysis, ROS scavenging, and cell organization. In addition, millimeter-wave irradiation may promote tolerance against flooding through the regulation of auxin contents in wheat.

## Figures and Tables

**Figure 1 ijms-23-10360-f001:**
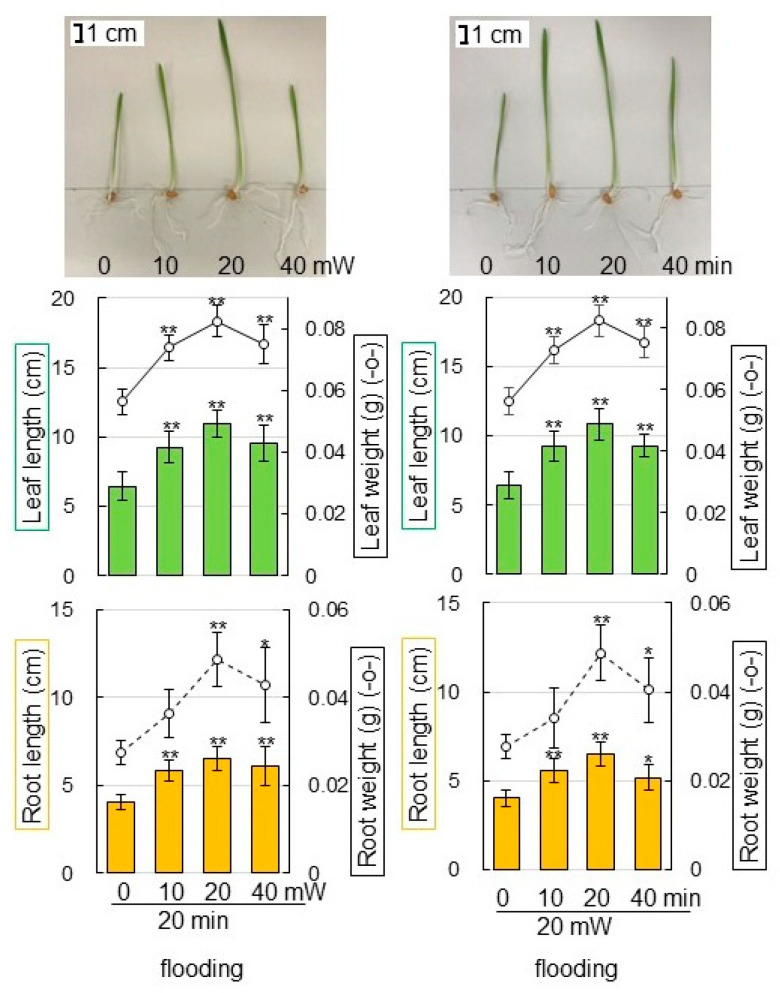
Morphological effect of millimeter-wave irradiation with the various power and time on wheat under flooding stress. Wheat seeds were irradiated with 0, 10, 20, and 40 mW of millimeter waves for 0, 10, 20, and 40 min and sowed (Appendix A). For non-flooded group, samples were collected at 6 days after sowing (Appendix A). For flooded group, wheat seedlings were treated with 3-day flooding after 3-day germination and samples were collected. Leaf length (green column), leaf-fresh weight (black solid line graph), main-root length (orange column), and total-root fresh weight (black dotted line graph) were measured as morphological parameters. Bar indicates 1 cm. The data are given as the mean ± SD from three independent biological replicates. Asterisks indicate significant changes of irradiated groups compared with unirradiated group according to Student’s *t*-test (** *p* < 0.01; * *p* < 0.05).

**Figure 2 ijms-23-10360-f002:**
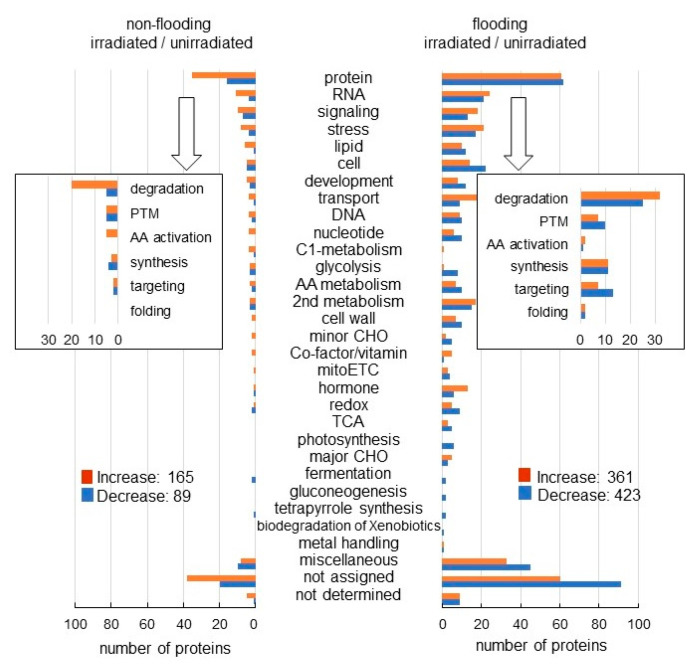
Functional categories of proteins with differential abundance in wheat irradiated with millimeter waves with or without flooding stress. Wheat seeds were irradiated without (unirradiated) or with (irradiated) 20 mW of millimeter waves for 20 min and exposed without (non-flooded) or with (flooded) flooding stress. Non-flooded and flooded samples were collected from both the unirradiated group and the irradiated group. After proteomic analysis, functional categories of significantly changed proteins (*p* < 0.05) from irradiated/unirradiated wheat without or with flooding stress were determined using MapMan bin codes (Appendix A). Orange color indicates “increased proteins” and blue color indicates “decreased proteins” in irrigated wheat compared with unirradiated wheat. Abbreviations: mitoETC, mitochondrial-electron transport chain; TCA, tricarboxylic-acid cycle; PTM, post-translational modification; and AA, amino acids. “not assigned” indicates proteins without ontology or characterized functions.

**Figure 3 ijms-23-10360-f003:**
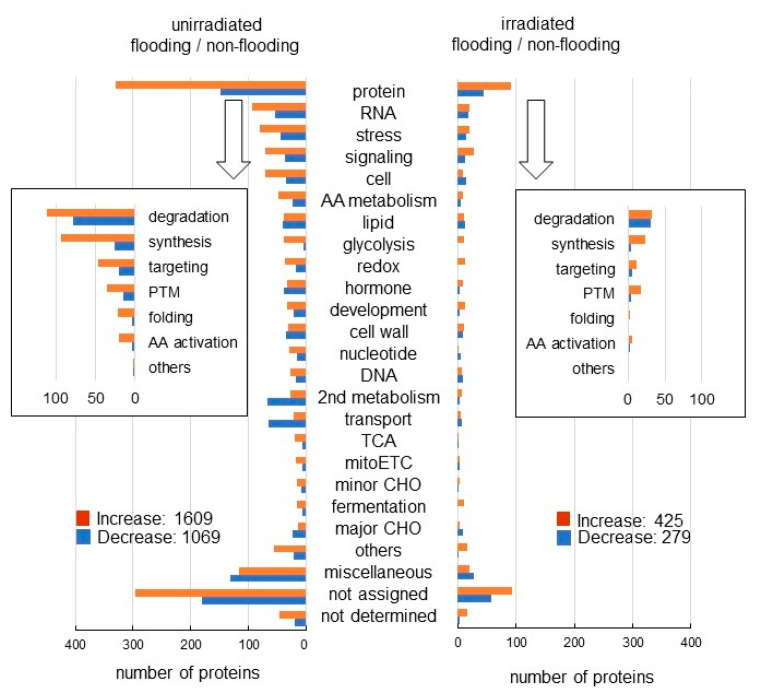
Functional categories of proteins with differential abundance in wheat irradiated with or without millimeter waves under flooding stress. Wheat seeds were irradiated without (unirradiated) or with (irradiated) 20 mW millimeter waves for 20 min and exposed without (non-flooded) or with (flooded) flooding stress. Non-flooded and flooded samples were collected from both unirradiated group and irradiated group. After proteomic analysis, functional categories of significantly changed proteins (*p* < 0.05) from flooding/non-flooding wheat without or with irradiation were determined using MapMan bin codes (Appendix A). Orange color indicates “increased proteins” and blue color indicates “decreased proteins” under flooding stress compared with non-flooding stress. Abbreviations: mitoETC, mitochondrial electron transport chain; TCA, tricarboxylic acid cycle; PTM, post-translational modification; and AA, amino acids. “not assigned” indicates proteins without ontology or characterized functions.

**Figure 4 ijms-23-10360-f004:**
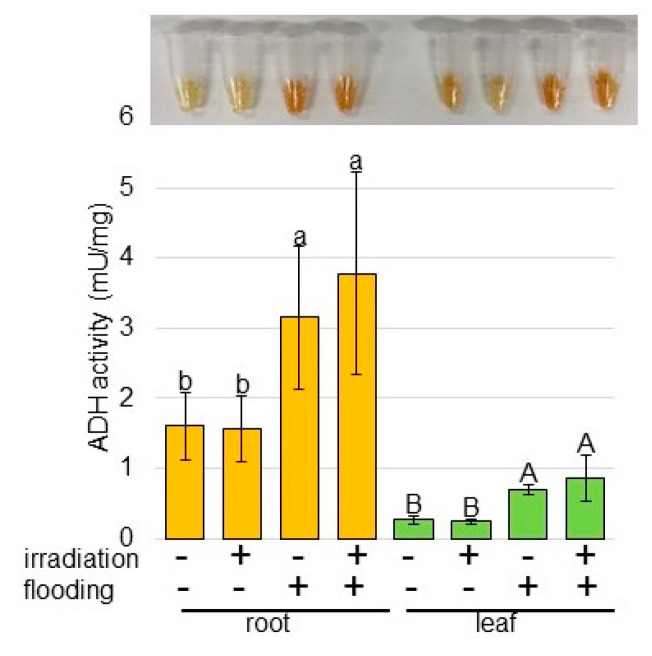
Activity assay of ADH involved in fermentation in wheat irradiated with millimeter waves under flooding stress. Wheat seeds were irradiated with (+) or without (−) 20 mW millimeter waves for 20 min and exposed with (+) or without (−) flooding stress. ADH activity was analyzed in protein extracted from root and leaf of wheat. The data are given as the mean ± SD from three independent biological replicates. Mean values in each point with different letters are significantly different according to one-way ANOVA followed by Tukey’s multiple comparisons (*p* < 0.05).

**Figure 5 ijms-23-10360-f005:**
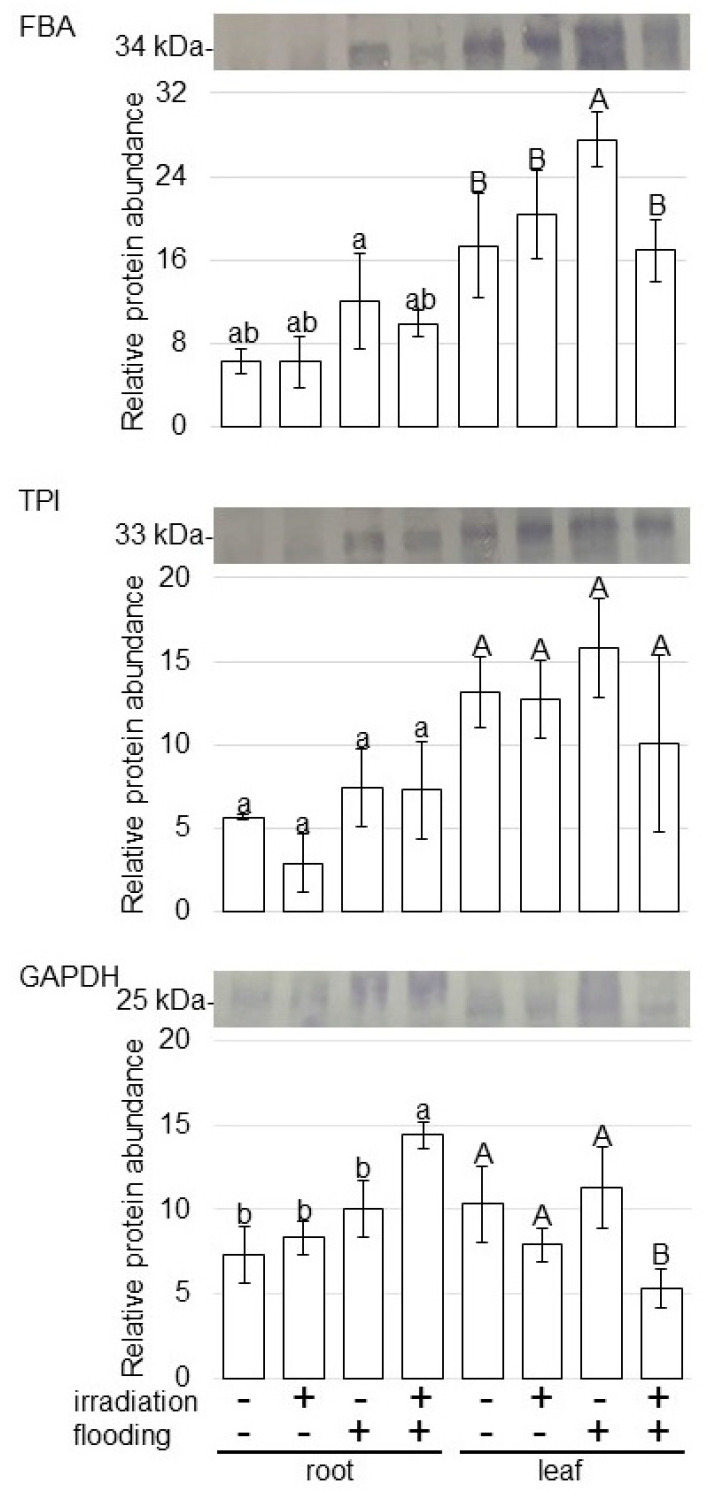
Immunoblot analysis of proteins involved in glycolysis in wheat irradiated with millimeter waves under flooding stress. Proteins extracted from the roots and leaves of wheat seedlings were separated on SDS-polyacrylamide gel by electrophoresis and transferred onto membranes. The membranes were cross-reacted with anti-FBA, TPI, and GAPDH antibodies. Staining pattern with Coomassie-brilliant blue was used as a loading control (Appendix A). The integrated densities of bands were calculated using ImageJ software. The data are given as the mean ± SD from three independent biological replicates (Appendix A). The statistical analysis is same as Figure 4.

**Figure 6 ijms-23-10360-f006:**
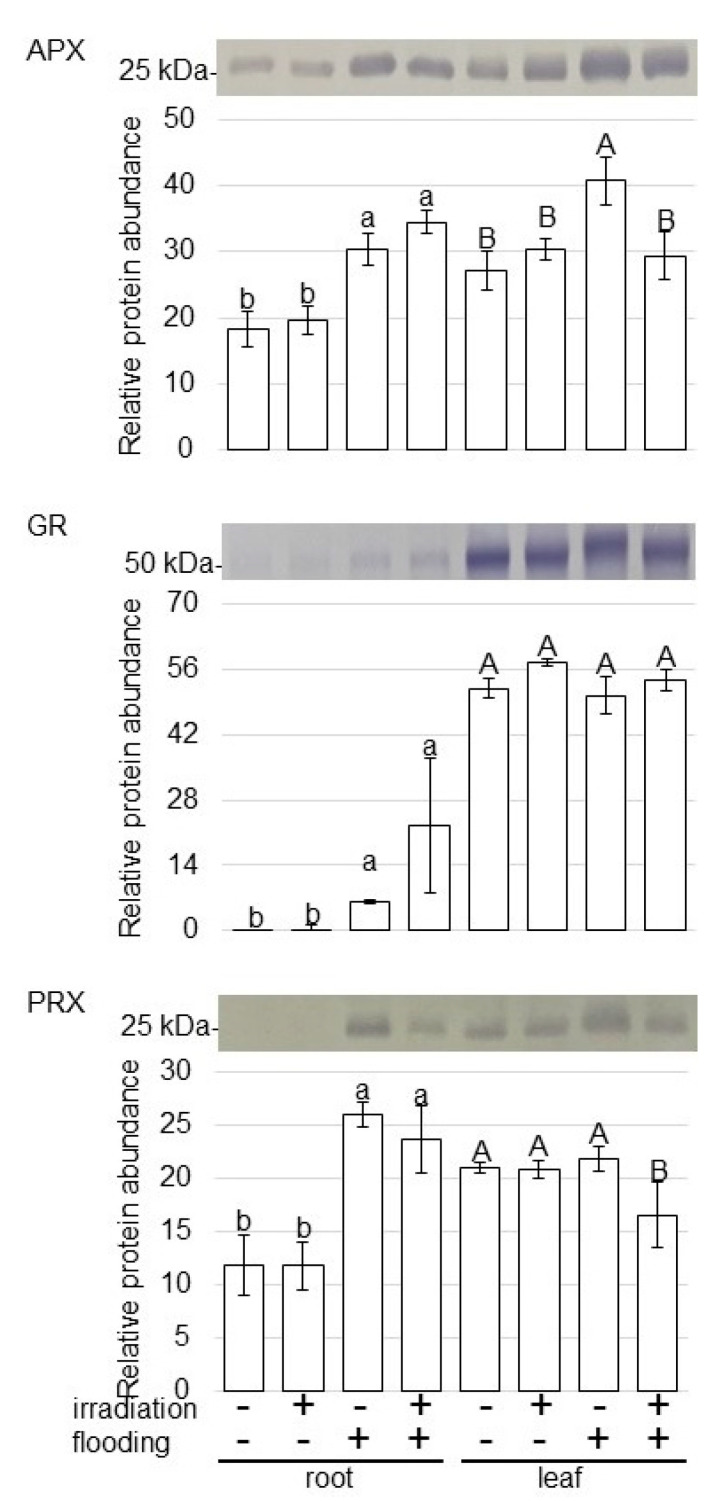
Immunoblot analysis of proteins involved in ROS scavenging in wheat irradiated with millimeter waves under flooding stress. Proteins extracted from roots and leaves of wheat seedlings were separated on SDS-polyacrylamide gel by electrophoresis and transferred onto membranes. The membranes were cross-reacted with anti-APX, GR, and PRX antibodies. Staining pattern with Coomassie-brilliant blue was used as a loading control (Appendix A). The integrated densities of bands were calculated using ImageJ software. The data are given as the mean ± SD from three independent biological replicates (Appendix A). The statistical analysis is same as Figure 4.

**Figure 7 ijms-23-10360-f007:**
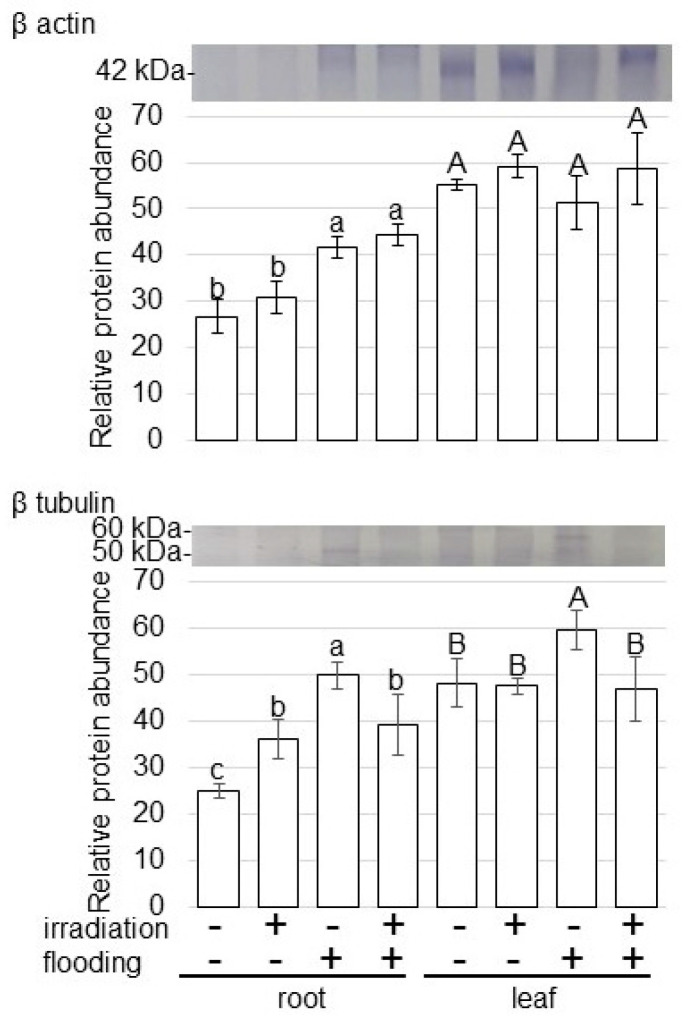
Immunoblot analysis of proteins involved in cell organization in wheat irradiated with millimeter waves under flooding stress. Proteins extracted from root and leaf of wheat seedlings were separated on SDS-polyacrylamide gel by electrophoresis and transferred onto membranes. The membranes were cross-reacted with anti-β actin and β tubulin antibodies. Staining pattern with Coomassie-brilliant blue was used as loading control (Appendix A). The integrated densities of bands were calculated using ImageJ software. The data are given as the mean ± SD from three independent biological replicates (Appendix A). The statistical analysis is same as Figure 4.

**Figure 8 ijms-23-10360-f008:**
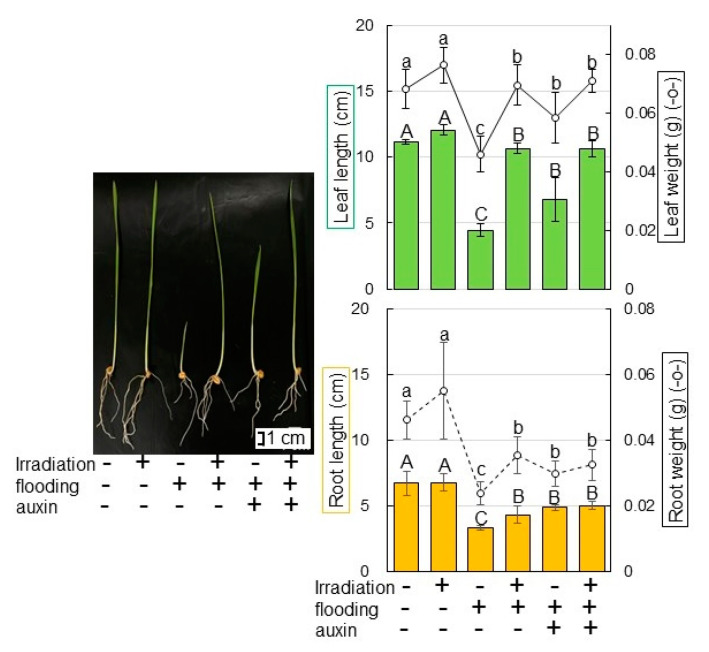
Morphological effect of auxin application on wheat under flooding stress. For the non-flooded group, samples were collected at 6 days after sowing. For flooded group, 3-day-old wheat seedlings were treated with 3-day flooding and samples were collected. For auxin treated group, 3-day-old wheat seedlings were treated with auxin under flooding stress for 3 days and samples were collected. Leaf length (green column), leaf-fresh weight (black solid line graph), main-root length (orange column), and total-root weight (black dotted line graph) were measured as morphological parameters. Bar indicates 1 cm. The data are given as the mean ± SD from three independent biological replicates. The statistical analysis is same as Figure 4.

## Data Availability

For MS data, RAW data, peak lists, and result files have been deposited in the ProteomeXchange Consortium [53], via the jPOST [54], partner repository under data-set identifiers PXD031669.

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
