# Peer review of "Proteomic and Biochemical Approaches Elucidate the Role of Millimeter-Wave Irradiation in Wheat Growth under Flooding Stress"

_ijms, 2022, doi:10.3390/ijms231810360_

Round 1

Reviewer 1 Report

This work is a continuation of the proteomic research of the authors on the effect of millimeter-wave irradiation of seeds as a convenient treatment to enhance stress resilience of crops, analyzing roots as the primary plant part which suffers from flooding stress. The previous papers focused on chickpea and soybean. The ms is written clearly and concisely in a good English. Proteomic data are supported with morphological and biochemical analyses; in addition, auxin application is used to support some conclusions based on proteomic findings. I found only minor technical mistakes to be corrected such as tables S4 and S5 – same title 9but in the text all is correct), figures S1 and S2 do not show morphological changes but introduce the milimeter-wave source and the experimental design.  This is an interesting and valuable piece of work.

Author Response

Reviewer 1

This work is a continuation of the proteomic research of the authors on the effect of millimeter-wave irradiation of seeds as a convenient treatment to enhance stress resilience of crops, analyzing roots as the primary plant part which suffers from flooding stress. The previous papers focused on chickpea and soybean. The ms is written clearly and concisely in a good English. Proteomic data are supported with morphological and biochemical analyses; in addition, auxin application is used to support some conclusions based on proteomic findings.

I found only minor technical mistakes to be corrected such as tables S4 and S5 – same title 9but in the text all is correct),

Answer: We are sorry we made a mistake. The titles of Table S4 and S5 have been corrected in the text and supplemental tables. Additionally, the explanations of Table S2 and S3 have been changed.

figures S1 and S2 do not show morphological changes but introduce the milimeter-wave source and the experimental design.  This is an interesting and valuable piece of work.

Answer: Thank you very much for pointing this out. Because Figure S1 and Figure S2 was the explanation of millimeter-wave source and the experimental design, an explanation of millimeter-wave source and the experimental design has been added using Figure S1 and Figure S2 in the section of “2.1 Morphological Changes of Wheat Irradiated with Millimeter Waves under Flooding Stress” in red color.  

Reviewer 2 Report

The work Komatsu et al., «Proteomic and Biochemical Approaches Elucidate the Role of  Millimeter-Wave Irradiation in Wheat Growth under Flooding  Stress»  submitted for review is relevant, since the study of safe methods for increasing the yield of the most important agricultural crop - wheat is very important. In the study, the authors analyzed the metabolic rearrangements of the most important components of the fundamental systems of plant vital activity - growth, photosynthesis, respiration and the antioxidant system. The authors selected the wavelength. which increases the resistance of wheat to the action of flooding. A lot of work has been done. In the course of reading the material, some questions and additions arise. I think they should be taken into account. This will make the material more understandable and reveal its value.

* important is the wheat variety, it may be worth giving its characteristics.

  • The lines in the Introduction 55-58 are the same as 271-273, in the discussion you can formulate the idea differently.
  • in the Results part, the article mentions S1,2,3,4,5 .. 10 this is not in the drawings, which is meant by Table S1, 2,3.4, Fig. S2. I ask you to specify where it is in the presented figures, I did not find it. Review captions.
  • In figures 2 and 3, the change in auxin metabolism refers to AA or hormones. In the method part, indicate where the concentration of 1 mg/L was taken from and why this particular form of the hormone. Undoubtedly the hormonal regulation of rearrangements during any influence on plants is the key, but in your case it is necessary to justify this more clearly. And give in the part in Introduction about this too.
  • In the Introduction, lines 42-45, there are two types of flooding, which type was used in your work. From lines 385-386 it is not entirely clear, it's just that there was always water on top and the above-ground part was in the air.
  •  the Discussion needs to be expanded. there is very little discussion of antioxidant enzymes, you only talk about APX ,and the additional work of GH (Fig. 6) is very important in realizing flood resistance.

Author Response

Reviewer 2

The work Komatsu et al., «Proteomic and Biochemical Approaches Elucidate the Role of  Millimeter-Wave Irradiation in Wheat Growth under Flooding  Stress»  submitted for review is relevant, since the study of safe methods for increasing the yield of the most important agricultural crop - wheat is very important. In the study, the authors analyzed the metabolic rearrangements of the most important components of the fundamental systems of plant vital activity - growth, photosynthesis, respiration and the antioxidant system. The authors selected the wavelength. which increases the resistance of wheat to the action of flooding. A lot of work has been done.

In the course of reading the material, some questions and additions arise. I think they should be taken into account. This will make the material more understandable and reveal its value.

1.important is the wheat variety, it may be worth giving its characteristics.

Answer: Thank you very much for your suggestion. This information has been added in the section “3.1. Millimeter-Wave Irradiation Has Positive Effect on Wheat Growth under Flooding Stress”. It is as follows: “In this study, the Japanese bread wheat cultivar Nourin 61 was used. Nourin 61 is a representative Japanese cultivar of bread wheat, which is characterized by broad adaptation and environmental robustness (Ishikawa, 2010). It was utilized for a broad range of physiological and molecular studies, such as mutant screening and transgenic experiments (Shimazu et al., 2021). It is indicated that the features of Nourin 61 might be established as the reference genotype for adaptation and breeding research.”

2.The lines in the Introduction 55-58 are the same as 271-273, in the discussion you can formulate the idea differently.

Answer: As suggested, these redundant explanations with the introduction have been deleted in the discussion.

3.in the Results part, the article mentions S1,2,3,4,5 .. 10 this is not in the drawings, which is meant by Table S1, 2,3.4, Fig. S2. I ask you to specify where it is in the presented figures, I did not find it. Review captions.

Answer: These supplemental materials have been already submitted in the system of Journal. I will ask the Editorial Assistant what is going on.

4.In figures 2 and 3, the change in auxin metabolism refers to AA or hormones. In the method part, indicate where the concentration of 1 mg/L was taken from and why this particular form of the hormone. Undoubtedly the hormonal regulation of rearrangements during any influence on plants is the key, but in your case it is necessary to justify this more clearly. And give in the part in Introduction about this too.

Answer: Thank you very much for pointing this out. Based on your comments, the following sentences have been added in the section “3.4. Millimeter-Wave Irradiation Regulates Auxin Metabolism in Wheat under Flooding Stress” as follows: “On the other hand, out of 4 auxins, which are indole-3-acetic acid, indole-3-propionic acid, indole-3-butyric acid, and 1-naphthylacetic acid, treatment with indole-3-butyric acid resulted in a high rooting rate and beneficial root morphology (Quan et al., 2022). Additionally, 1.5 mg/L indole-3-butyric acid exhibited the highest rooting responses and 1.0 mg/L indole-3-butyric acid improved fruit yield and biomass (Thorat et al., 2022). “. However, the explanation of auxin metabolism has been written in discussion section, because auxin metabolism was found by this research. We hope that reviewer might understand this correction.

5.In the Introduction, lines 42-45, there are two types of flooding, which type was used in your work. From lines 385-386 it is not entirely clear, it's just that there was always water on top and the above-ground part was in the air.

Answer: We are sorry for the lack of clarity. In this study, submergence was used. In the section of “4.1. Plant Material, Millimeter-Wave Irradiation, and Treatment “, the flooding condition has been corrected in red color. It is as follows: “To induce flooding stress, water was added to 5 cm above the sand surface to immerse 3-day-old wheats for 3 days.” And also, in the section “1. Introduction”, it has been clarified.

6.the Discussion needs to be expanded. there is very little discussion of antioxidant enzymes, you only talk about APX ,and the additional work of GH (Fig. 6) is very important in realizing flood resistance.

Answer: Thank you very much for your comments. Based on the results in this study, the section “3.3. Millimeter-Wave Irradiation Suppresses ROS Scavenging and Cell Organization in Wheat under Flooding Stress” has been corrected as follows: “In this study, among ROS-scavenging enzymes, APX increased in leaf under flooding and recovered to the control level in irradiated wheat (Figure 6). On the other hand, in wheat leaf, the abundance of GR and PRX did not change with flooding stress (Figure 6). In wheat root, the abundance of APX, GR, and PRX increased under flooding stress; however, this accumulation did not recover with millimeter-wave irradiation. These results indicate that APX via the regulation of ROS production could be considered an important component of adaptive responses to flooding for wheat irradiated with millimeter waves compared with other ROS-scavenging enzymes.”